# Field Evaluation of a Loop-Mediated Isothermal Amplification (LAMP) Platform for the Detection of *Schistosoma japonicum* Infection in *Oncomelania hupensis* Snails

**DOI:** 10.3390/tropicalmed3040124

**Published:** 2018-12-15

**Authors:** Zhi-Qiang Qin, Jing Xu, Ting Feng, Shan Lv, Ying-Jun Qian, Li-Juan Zhang, Yin-Long Li, Chao Lv, Robert Bergquist, Shi-Zhu Li, Xiao-Nong Zhou

**Affiliations:** 1Key Laboratory of Parasite and Vector Biology, Ministry of Health; National Institute of Parasitic Diseases, Chinese Center for Disease Control and Prevention, Shanghai 200025, China; qinzq1008@hotmail.com (Z.-Q.Q.); xujing@nipd.chinacdc.cn (J.X.); fengting@nipd.chinacdc.cn (T.F.); lvshan@nipd.chinacdc.cn (S.L.); yjqiancn@163.com (Y.-J.Q.); zhanglj@nipd.chinacdc.cn (L.-J.Z.); liyl@nipd.chinacdc.cn (Y.-L.L.); lvchao@nipd.chinacdc.cn (C.L.); lisz@chinacdc.cn (S.-Z.L.); 2Ingerod, SE-454 94 Brastad, Sweden; robert.bergquist@outlook.com

**Keywords:** *Schistosoma japonicum*, *Oncomelania hupensis*, snail, 28S ribosomal DNA, PCR, loop-mediated isothermal amplification (LAMP), pooled samples, China

## Abstract

*Schistosoma* infection in snails can be monitored by microscopy or indirectly by sentinel mice. As both these approaches can miss infections, more sensitive tests are needed, particularly in low-level transmission settings. In this study, loop-mediated isothermal amplification (LAMP) technique, designed to detect a specific 28S ribosomal *Schistosoma japonicum* (Sj28S) gene with high sensitivity, was compared to microscopy using snail samples from 51 areas endemic for schistosomiasis in five Chinese provinces. In addition, the results were compared with those from polymerase chain reaction (PCR) by adding DNA sequencing as a reference. The testing of pooled snail samples with the LAMP assay showed that a dilution factor of 1/50, i.e., one infected snail plus 49 non-infected ones, would still result in a positive reaction after the recommended number of amplification cycles. Testing a total of 232 pooled samples, emanating from 4006 snail specimens, showed a rate of infection of 6.5%, while traditional microscopy found only 0.4% positive samples in the same materials. Parallel PCR analysis confirmed the diagnostic accuracy of the LAMP assay, with DNA sequencing even giving LAMP a slight lead. Microscopy and the LAMP test were carried out at local schistosomiasis-control stations, demonstrating that the potential of the latter assay to serve as a point-of-care (POC) test with results available within 60–90 min, while the more complicated PCR test had to be carried out at the National Institute of Parasitic Diseases (NIPD) in Shanghai, China. In conclusion, LAMP was found to be clearly superior to microscopy and as good as, or better than, PCR. As it can be used under field conditions and requires less time than other techniques, LAMP testing would improve and accelerate schistosomiasis control.

## 1. Introduction

Schistosomiasis, one of the neglected tropical diseases (NTDs), is a public health problem caused by one of six species of the *Schistosoma* parasite that affects >200 million people in Africa, South America and Southeast Asia. However, the prevalence of schistosomiasis, based on stool examination and urine filtration, strongly understates the presumed real figure, as indicated by more sensitive techniques [1,2]. In China, the Philippines and three small pockets of the Indonesian island Sulawesi, the disease is a zoonosis caused by *Schistosoma japonicum* [1,2,3]. In China, the required intermediate snail host, *Oncomelania hupensis*, is widely distributed in the country’s endemic areas, ranging from the Yangtze River Valley and the southern plains, to the mountainous regions of Sichuan and Yunnan in the west [4]. In 2004, a new integrated control strategy was introduced [5,6] involving reduction of infection sources by fencing off transmission sites, the replacement of water buffaloes (an important reservoir) with tractors for agricultural work and improved sanitation via access to clean water and latrines. These approaches have markedly reduced the infection rate in humans, domestic animals and the intermediate snail host [7]. The changes accomplished are so profound that it has become difficult to monitor the remaining transmission sites by only testing humans and domestic animals [8]. Conversely, snail diagnosis by microscopy or the use of the sentinel mice approach [9] are not only labor-intensive and time-consuming, but are also not sufficiently sensitive. To sustain the success achieved with regard to schistosomiasis elimination in China, highly sensitive snail-monitoring systems capable of assessing residual transmission in real time are now needed.

Due to its high sensitivity, molecular diagnosis has emerged as a promising approach for the detection of a suspected, low-level presence of pathogens [10,11,12,13]. However, a lack of the resources essential for this kind of diagnostics—such as bio-safety cabinets, a stable supply of electricity and well-experienced technicians, which are rare in peripheral laboratories in developing countries—limits the implementation of sophisticated technology.

The loop-mediated isothermal amplification (LAMP) technology employs a polymerase that amplifies the target DNA gene sequence with high specificity and rapidity under isothermal conditions [14]. Hamburger and colleagues investigated the use of this technique for the detection of infections due to *S. mansoni* and *S. haematobium,* showing excellent results, not only confirming that the LAMP technique works in the laboratory, but also in the field in Africa [15]. The usefulness and high sensitivity of LAMP-assisted snail diagnosis was later confirmed in Brazil by Gandasegui et al. [16] and in China by Kumagai et al. [17]. The latter research group developed a diagnostic platform based on a target 28S ribosomal DNA (rDNA) specific for *S. japonicum* (not reacting with *S. mansoni*) and showed that snails experimentally infected with only one miracidium could be detected less than 24 h after infection. As part of this study, we explored the application of LAMP using pooled snail samples, i.e., instead of testing each snail separately, we combined snails, however never in numbers exceeding 50 snails per pooled sample, based on preliminary dilution tests (see Results). The protocol used was derived from a visual LAMP-detection method developed by Tomita et al. [18], where the amplification of the pyrophosphate ion by-product combines with a divalent metal ion to form an insoluble salt. Adding calcein, a bivalent fluorescein/manganese complex, to the reaction solution results in a strong fluorescent signal from positive reactions, enabling visual discrimination by the naked eye without specialized equipment. We compared the results of the LAMP assay with the outcome using a polymerase chain reaction (PCR), applying DNA sequencing to determine the identity of the amplified product.

The purpose of the present study was not only to confirm the sensitivity of the LAMP method when used for the detection of *S. japonicum* DNA in infected snails in known endemic settings, but also to investigate and validate its application under field conditions soon after snail collection. An added aim was to determine if the LAMP procedure could be speeded up by testing pooled DNA samples in place of testing the snails individually.

## 2. Materials and Methods

This study constitutes an evaluation of LAMP-based snail diagnosis with the eventual aim of being part of a platform integrating different kinds of data and, thereby, enabling improved surveillance of schistosomiasis transmission. *O. hupensis* snails were collected during the spring over a period of three years (2013–2015) from all endemic regions in China, covering five provinces. The LAMP approach used is shown schematically in the form of a flow chart (Figure 1). As we were interested to see if the field work could be accelerated by testing samples consisting of pooled snails, DNA from various numbers of snails was investigated before the main study was undertaken.

### 2.1. Ethical Statement

Although this study did not include human sera or experimental animal data, an ethical statement is required to initiate any research project. Thus, we hereby state that all procedures performed within this study were conducted following animal husbandry guidelines of the Chinese Academy of Medical Sciences and with permission from the Experimental Animal Committee (National Institute of Parasitic Diseases (NIPD), Chinese Center for Diseases Control and Prevention, (China CDC) with ethical clearance number IPD-2012-5.

### 2.2. Snail Sampling Procedure

*O. hupensis* snails were collected from areas with known human schistosomiasis infection rates varying between 0.9% and 2.8% [19]. We used snails collected in April/May each year from 51 villages in 15 counties in Hunan, Hubei, Jiangxi, Anhui and Yunnan provinces. Most collection spots were in marshlands, except in Yunnan Province, where the snails were found in irrigation and drainage ditches. Systematic sampling was applied for snail investigation using survey frames of 250 m^2^ in the marshlands and smaller areas spaced 50 m apart along the ditches [20,21]. All live snails found, together numbering 4006 specimens, were kept and transferred to the laboratory for testing.

### 2.3. Microscopy Testing and DNA Extraction

After having been crushed by pressure between clean glass plates and the pieces of shell removed, the snails were examined individually under the microscope at low magnification (generally 10×) to certify whether cercariae and/or sporocysts were present. After microscopy testing, 232 pooled snail samples from the 4006 single snails, were collected for further genomic DNA extraction. The snail soft tissues were pooled in clean 2.0 or 10.0 mL tubes (not exceeding 50 snail bodies in each) with 400–1000 μL of DNA lysis buffer (Qiagen, Valencia, CA, USA) added into each tube and the tissue collections homogenized. Subsequently, 20–50 μL of ProteinK (Qiagen, Valencia, CA, USA) was added to the snail homogenates with the tubes kept in a water bath at 56 °C, for 2–3 h, after which the supernatant of snail homogenates was saved and passed through a DNA binding column (Qiagen, Valencia, CA, USA). Finally, the dedicated DNA binding column was washed by the washing buffer (Qiagen, Valencia, CA, USA) and the genomic DNA recovered by eluting the DNA-binding column with nuclease-free water. The recovered DNA samples were used for further testing by the LAMP assay and conventional PCR.

### 2.4. PCR Assay

A conventional PCR assay (PCR kit purchased from Takara Biotech, Dalian, China) was carried out in parallel with the LAMP assay (see below). Paired primer strands, i.e., 5′-GGTTTGACTATTATTGTTGAGC-3′ and 5′-CTCACCTTAGTTCGGACTGA-3′ targeting the *S. japonicum* 28S (Sj28S) rDNA (Accession:EU835689.1) were used [17]. The ingredients were 1 μL of each primer (10 pmol/L each), 3 μL DNA template, 0.2 mM deoxynucleotide (dNTP) solution, 1.25 units of highly thermostable DNA polymerase from the thermophilic bacterium *Thermus aquaticus* (Taq DNA) and 2.5 μL 10× buffer (pH 8.8 Tris-HCl, KCl and MgCl_2_). The initial activation (cycle 1) was set for 3 min at 94 °C followed by 33 cycles of 30 s at 94 °C, 56 °C for 50 s and 72 °C for 1 min. The final extension step was carried out for 7 min at 72 °C. The amplified product was visualized by agarose gel electrophoresis (1.5%) with ethidium bromide staining.

### 2.5. DNA Sequencing

In order to make sure that the PCR product of all 14 positive samples exactly matched the target sequence, the 330 bp band of each PCR product was separated by gel electrophoresis, cloned into the Pmd19-T vector (Takara Biotech, Dalian, China), transferred into *E. coli* cells, strain DH5α, and then cultured in Luria-Bertani (LB) medium with ampicillin (100 μg/mL) at 37 °C in a 5.0 mL tube. After 12–16 h of culture, plasmid DNA was extracted from the bacterial colonies using a DNA extraction kit. The purified plasmid DNA was sent to a commercial company (Sangon Biotech, Co., Ltd., Shanghai, China) for sequence analysis, confirming the sample to be a complete match to Sj28S rDNA when compared with the National Center for Biotechnology Information (NCBI) database. One sample found to be positive by LAMP but negative by PCR, was subjected to an additional agarose gel analysis, thus we isolated the target band and sent the purified product for DNA sequence analysis to Sangon Biotech.

### 2.6. LAMP Assay

(A) Reagent stock solution set-up and primer sequence

The 2× reaction mixture: pH 8.8 Tris buffer (40 mM), KCl (20 M), MgSO_4_ (16 mM), (NH_4_)_2_SO_4_ (20 mM), Tween 20 (0.2%), betaine (1.6 M), dNTPs (2.8 mM each).Calcein working solution (2 × composition): Calcein (50 μM), MnCl_2_ (1 mM).Sj28S gene primers [17], (5′-3′, Sangon; HPLC purification): F3 (GCTTTGTCCTTCGGGCATTA), B3 (GGTTTCGTAACGCCCAATGA), FIP (ACGCAACTGCCAACGTGACATACTGGTCGGCTTGTTACTAGC), BIP (TGGTAGACGATCCACCTGACCCCTCGCGCACATGTTAAACTC)

(B) Amplification (60–90 min)

We produced a reaction mixture of 7.5 μL nuclease-free water, 12.5 μL of 2× reaction buffer, 1.0 μL of 25× primer mixture, 1.0 μL (8 μ/μL) of Bst, a *Bacillus stearothermophilus* DNA polymerase homologue used for DNA strand displacement and 1.0 μL calcein. We used 0.2 mL reaction tubes dispensing 23 μL of the reaction mixture together with 2.0 μL of the sample to be tested into each tube. In addition, 23 μL of the reaction mixture was added to the Sj28S rDNA target in a second tube (the positive control) and also to the nuclease-free water in second tube (the negative control). All reaction tubes were incubated at 65 °C for 60~90 min, followed by the inactivation of the enzyme by the incubation of the tubes at 85 °C for 5 min. The tubes were then observed by the unaided eye to observe the color of the reaction changing from orange to yellow-green in the presence of LAMP (see Results).

#### 2.6.1. Testing Samples with Life Cycle Stages of *S. japonicum*

Laboratory-bred *O. hupensis* snails were challenged with *S. japonicum* miracidia and the next life cycle stages of *S. japonicum* were thus transferred to the *O. hupensis* snails. To confirm whether DNA from different developing stage of *S. japonicum* could be detected by the LAMP assay, we performed the test with DNA collected from a mother sporocyst, a daughter sporocyst and a mature cercaria from these laboratory-based snails (infected and non-infected). Furthermore, to study whether DNA from other *Schistosoma* species would cross-react in the LAMP assay *Trichobilharzia* cercariae, emanating from a common bird-specific *Schistosoma* species, was also subjected to testing using the same procedure as described above under 2.5 and 2.6.

#### 2.6.2. Pooled Snail Samples

To evaluate the detection limit of the LAMP method before the main study, we performed a preliminary test in which one schistosome-positive snail was mixed with different numbers of schistosome-negative snails at the following ratios: 1/4, 1/9, 1/19, 1/49, 1/99 and 1/199. For practical reasons, each snail pool in the main study came from the same area (though the different pools investigated contained different sets of snails). Depending on availability, the number of snails in the pools could not be standardized but it was always kept at ≤50. The result of this exercise was 232 snail pools made out of the total number of snails collected (4006). Depending on how many snails were pooled, the pools were kept in 2.0 mL or 10 mL centrifuge tubes together with 400 μL or 1.0 mL, of a lysis buffer from a DNeasy blood and tissue kit, (Qiagen, Valencia, CA, USA).

### 2.7. Validation

To assess the quality of the LAMP assay, a panel of nucleic acid extracts and test kits were dispatched to 28 separate testing health agency laboratories in the endemic areas at the provincial and county levels, where laboratory personnel were well-trained to operate the LAMP assay. Each nucleic acid sample was thawed, divided into 100 μL aliquots, coded and refrozen. The next day, all samples were packaged and sent by overnight shipment to the 28 separate laboratories. Each laboratory tested each sample 5 times using a standard LAMP protocol provided by National Institute of Parasitic Diseases (NIPD) at China CDC, based in Shanghai. The samples were validated against the true results, kept at the NIPD, which were unknown to the staff carrying out the testing in the health agency laboratories. The outcomes were scored according to a 0–100 scale, where 5 equal results were given a score of 100, 4 equal results were given a score of 80, and so on. To make sure that the LAMP test had the required specificity, it was applied (in parallel with the PCR assay) to 20 pools produced from a total of 1000 individual snails that had previously also been investigated by microscopy.

## 3. Results

As seen in Figure 2, the preliminary test of the snail pools in different ratios showed that large-scale testing could be carried out based on 50 snails without the risk of missing a positive result.

The ready-to-use LAMP test kit evaluated here detected 7.5 times more infected snails than microscopy, while the PCR results were consistent with those of the LAMP assay, except with respect to a single snail pool that was found to be positive by LAMP and negative by PCR. Out of the 232 pooled snail samples tested, 217 were found to be negative and 14 positive by both assays (Table 1), while the remaining single sample, found positive by LAMP but negative by PCR, was found to be positive by DNA sequence analysis, underlining LAMP’s superiority. As can be seen in Table 2, none of the diagnostic approaches produced false positive results when applied to snail samples from non-endemic areas. The results further demonstrated that the schistosome DNA included in a single miracidium is sufficient to be amplified by the LAMP assay, making it possible to detect the schistosome infection already at the sporocyst stages in the snail, as well as when mature cercariae are ready for release. The *Trichobilharzia* cercariae used to test the specificity were negative, confirming the specificity of the test (Figure 3).

The 28 different laboratories at the provincial and county health agency levels, belonging to the inter-laboratory panel, demonstrated an almost total agreement with respect to the results over the three reported years. Only four laboratories showed slightly lower scores, one with a score of 80 in 2013, two with a score of 60 in 2014, and one with a score of 80 in 2014 and a score of 90 in 2015 (Table 3). Importantly, with the exception of one laboratory, those with scores lower than 100 in one year did not have the same score in other years.

## 4. Discussion

Estimates of the prevalence of *S. japonicum* infection in its obligatory vector snail can be used as a proxy for areas at risk of schistosomiasis. As already pointed out by Hamburger et al. [12] and reiterated by Abbasi et al. [13], the snail infection rate provides a measure of the transmission from humans (and other definitive reservoir hosts) to the snail and can, therefore, serve as a marker of residual infection in an area. However, traditional snail diagnosis depends on the labor-intensive, time-consuming individual dissection of thousands of collected snails from the field and the results are not sufficiently sensitive to support this statement, but with the advent of molecular diagnostics this has changed. The successful use of the LAMP technique shown here promises to revolutionize snail surveillance, not only in the laboratory but also in field [13,14,15,16,17]. The reported results are timely, as the use of snail control as a complementary approach along with chemotherapy is again being proposed as a potentially necessary means to achieve the elimination of schistosomiasis [22].

PCR-based assays have been around since the mid-1970s [10], but were not used for schistosomiasis diagnosis until the end of the 1990s and the early 2000s, first for snails [12] and later also for human infections [23,24]. However, the instability and variability inherent in enzymatic processes, as well as the need for advanced equipment, limit PCR applications in the field. In this regard, the LAMP approach is superior, as it can be adapted for application in field laboratories [25] by using ready-mixed reagents suitable for shipment at ambient temperature, together with sample storage under minimal refrigeration [13,14,15,16]. The development of a surveillance platform based on molecular diagnostic techniques and characterized by simplicity and reliability, yet with high throughput, requires an approach that is adaptable to county-level laboratories with limited resources. The manganese/calcein method of Tomita et al. [18] is an important contribution in this regard as it enables the recognition of small quantities of DNA by means of a fluorescent signal emitted from the sample solution after amplification, and has been successfully used in China by Kumagai et al. [17]. Other attempts to increase readability and sensitivity include the use of various dyes, e.g., SYBR green (Singh et al., 2017) or hydroxyl napthol blue (Ali et al., 2017) which have also produced good results [26,27]. Dyes have the advantage of being independent of refrigeration. 

Any increase of the dilution factor would considerably accelerate the area needed to be tested. We found a dilution ratio of 1/50 to be useful (Figure 2), but if the number of infected snails decreases, as it is supposed to do with the elimination program in force, the risk of mistakenly declaring an area free of transmission increases if the snail pool contains too few snails. This could be counteracted by using a higher number of snails in the pool and increasing the time of amplification for the test, however, the risk for false positives would then rise.

The strength of the present study is not only that it further improves the potential of the LAMP test that has already proved successful for snail diagnosis [16,17], but also that it targets a gene specific for *S. japonicum* and provides validation leading to the reliable use of the pooled-snail approach piloted by Hamburger et al. for *S. mansoni* [15] and Tong et al. for *S. japonicum* [20]. A further advantage of the LAMP technique is that although it is highly technical, it is easy to perform in basic laboratory settings common in rural areas. It is also easy to learn, as shown by the excellent agreement over three years in the many places included in this study. In addition, while sporocysts and parasite germ balls are easily missed by traditional microscopic methods, the snails containing mother or daughter sporocysts were both positive the day after infection. Therefore, the LAMP assay is an appropriate method for the detection of pre-patent infections. As seen in Table 1, the outcome of testing with PCR and the LAMP approach largely agreed, indicating that the sensitivity of the two techniques is the same in practice. However, theoretically the latter has an advantage since the one sample reacting negatively by PCR was positive when tested with the LAMP test. Comparison with microscopy, on the other hand, clearly showed that molecular testing is superior and should be used in the future. The excellent agreement between all 28 field laboratories (Table 3) shows that we are now ready to change from microscopy to molecular testing, preferably using LAMP, as it lends itself to use in the field. It is suggested that this kind of snail testing be included together with the diagnostic testing of humans and domestic animals in a joint surveillance and response platform based on only high-resolution techniques.

The risk of schistosomiasis still exists in China and snail control remains a significant challenge in the field [28]. In the present study, the LAMP results indicate that Hunan, Hubei, Jiangxi, Anhui, and Yunnan Provinces contain infected snails, underlining the risk of schistosomiasis transmission. The results can be used to guide further local investigations and snail control activities.

## 5. Conclusions

The LAMP platform is an effective method for monitoring snails in endemic field sites. Despite expensive reagents and the risk of contamination that requires specific training of the staff in charge, we recommend that the LAMP-based test replace microscopy for snail diagnosis due to its greater accuracy and reduced delay in delivering results.

## Figures and Tables

**Figure 1 tropicalmed-03-00124-f001:**
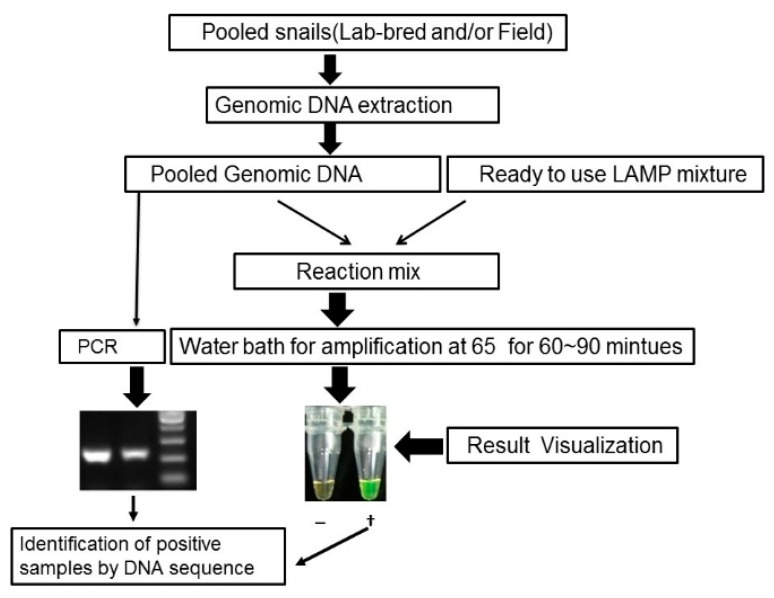
Flow chart of loop-mediated isothermal amplification (LAMP) and PCR for detecting *S. japonicum*-infected snails.

**Figure 2 tropicalmed-03-00124-f002:**
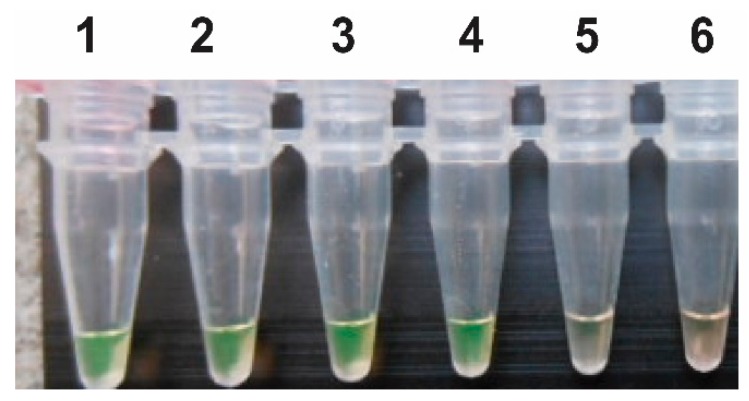
Investigation of the LAMP diagnostic capability by testing serial dilutions of the Sj28S gene component. The following dilutions of infected and uninfected snails are shown: 1/4 (tube no. 1); 1/9 (tube no. 2); 1/19 (tube no. 3); 1/49 (tube no. 4); 1/99 (tube no. 5) and 1/199 (tube no. 6).

**Figure 3 tropicalmed-03-00124-f003:**
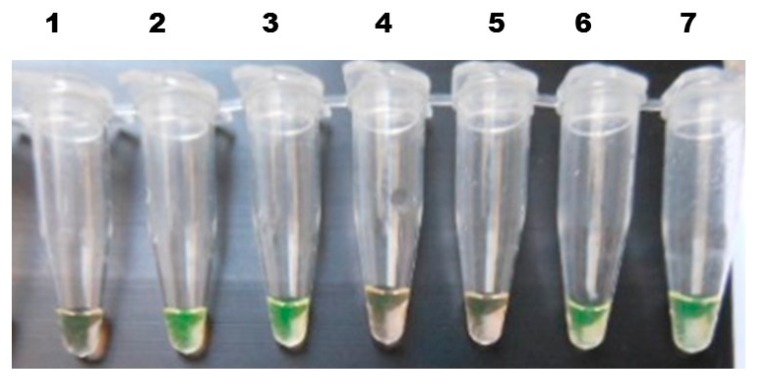
Detection of the Sj28S gene component in the different parasite developmental stages of *S. japonicum* in *O. hupensis* by the LAMP method. 1. Negative control (nuclease-free water); 2. Positive control (Sj28 plasmid DNA); 3. Cercariae; 4. Specificity control (*Trichobilharzia* cercariae); 5. Negative control (pooled snail DNA from a non-endemic area); 6. Mother sporocyst; 7. Daughter sporocyst.

**Table 1 tropicalmed-03-00124-t001:** Comparison between microscopy, LAMP and PCR in screening snail samples.

Province	No. of Counties Included	No. of Villages Included	No. of Snails Tested	No. of Pooled Samples	Microscopy		LAMP		PCR	
					Pos. *	%	Pos. *	%	Pos. *	%
Hubei	2	5	599	38	1	0.5	3	7. 9	3	7.9
Hunan	2	6	716	80	0	0	2	2.5	2	2.5
Jiangxi	6	25	1183	34	0	0	5	14.7	4	11.8
Anhui	2	6	698	43	0	0	1	2.3	1	2.3
Yunnan	3	9	810	37	0	0	4	10.8	4	10.8
**Total**	**15**	**51**	**4006**	**232**	**1**	**0.4**	**15**	**6.5**	**14**	**6.0**

* Positive.

**Table 2 tropicalmed-03-00124-t002:** Screening snail samples in three non-endemic areas: comparison between microscopy, LAMP and PCR.

Province	No. of Counties Included	No. of Villages Included	No. of Snails Tested	No. of Pooled Samples	Microscopy		LAMP		PCR	
					Pos. *	%	Pos. *	%	Pos. *	%
Shanghai	2	2	200	4	0	0	0	0	0	0
Zhejiang	2	2	500	10	0	0	0	0	0	0
Guangxi	2	2	300	6	0	0	0	0	0	0
**Total**	**6**	**6**	**1000**	**20**	**0**	**0**	**0**	**0**	**0**	**0**

* Positive.

**Table 3 tropicalmed-03-00124-t003:** Inter-laboratory comparison using LAMP for the detection of intermediate snail hosts infected by *S. japonicum.*

Province	Laboratory	Score 2013	Score 2014	Score 2015
Hunan	IPD	100	100	100
	Hanshou	-	100	100
	Yuanjiang	-	100	100
	Yueyang	-	100	100
Hubei	CDC	100	100	100
	Gongan	-	100	100
	Hanchuan	-	100	100
	Jiangling	-	100	100
Anhui	IPD	100	100	100
	Wuhu	-	100	100
	Anqin	-	100	100
	Guichi	-	100	100
Jiangxi	IPD	100	100	100
	Poyang	-	60	100
	Duchang	-	60	100
Jiangsu	IPD	100	100	100
	Qixia	-	100	100
Sichuan	CDC	100	100	100
	Renshou	-	100	100
	Guanghan	-	100	100
Yunnan	CDC	100	100	100
	Dali	-	80	90
	Eryuan	-	100	100
Shanghai	CDC	100	100	100
Guangdong	CDC	100	100	100
Fujian	CDC	100	100	100
Zhejiang	IPD	100	100	100
Guangxi	CDC	80	100	100

CDC = Center for Disease Control and Prevention; IPD= Provincial Institute of Parasitic Diseases; Scores are the levels of agreement between five tests. The number of samples from 2013 was not sufficient to be tested in all laboratories (signified with a dash in the table).

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
