# Peer review of "Field Evaluation of a Loop-Mediated Isothermal Amplification (LAMP) Platform for the Detection of Schistosoma japonicum Infection in Oncomelania hupensis Snails"

_tropicalmed, 2018, doi:10.3390/tropicalmed3040124_

Round 1

Reviewer 1 Report

All of the comments and edits are marked on the PDF file.

Author Response

Response to Reviewer 1 Comments

Dear Professor,

We appreciate you a lot for your nice comments.

We did a thorough revision of the manuscript, and addressed your comments point by point as followings.

Thanks and regards,

Zhiqiang Qin

Point 1: in Line 21 The particular snail species has been evaluated should be mentioned here.

Response 1:

Yes, the particular snail species were Oncomelania hupensis that have been evaluated.

Point 2: Line 44-47 This sentence needs revision. 1. The distribution is limited, not worldwide. 2. Current estimate is close to 290million prevalence according to GBD 2013 study.

Response 2:

Thanks. We deleted 'worldwide'. However, the prevalence is a sticky question as it varies constantly and GBD 2013 has already been supplanted by GBD 2017. We think >200 million is not wrong and it is stated by highly reliable authors. We keep the sentence as it is and added a follow-up sentence as follows: 'This prevalence, based on stool examination and urine filtration, strongly understates the real figure, which has been shown when more sensitive techniques are applied (Assaré, RK.: Tra, MBI.: Ouattara, M.: Hürlimann, E.: Coulibaly, JT.: N'Goran, EK.: Utzinger, J. Sensitivity of the Point-of-Care Circulating Cathodic Antigen Urine Cassette Test for Diagnosis of Schistosoma mansoni in Low-Endemicity Settings in Côte d'Ivoire. Am. J. Trop. Med. Hyg. 2018, doi:10.4269/ajtmh.18-0550.).This change of references can be done without changing numbers since we deleted Colley et al (now ref 2) and call the above ref this ref. number.

Point 3:Line 48-52 Revise this sentence to make the meaning clear. May be start with." In China, the intermediate snail host, Oncomelania hupensis is distributed..."

Response 3:

We put the line as follows: 'In China, the required intermediate snail host, Oncomelania hupensis, is widely distributed including the country's endemic areas that range from the Yangtze River Valley and the southern plains to the mountainous regions of Sichuan and Yunnan Provinces in the West [4].'

Point 4: Line 61 Any reference for this statement.

Response 4:

Yes, we use ref. 13 here.

Point 5: Line 69 Change with "implementation".

Response 5:

Thanks, we changed to 'implementation'.

Point 6: Line 70-80 This whole section needs to be broken down into small sentences to convey the message.

Response 6:

Change to 'The loop-mediated isothermal amplification (LAMP) technology employs a polymerase that amplifies the target DNA gene sequence with high specificity and rapidity under isothermal conditions [14]. Hamburger and colleagues investigated the use of this technique for the detection of infections due to S. mansoni and S. haematobium showing excellent results, not only confirming that the LAMP technique works in the laboratory, but also in the field in Africa [15]. The usefulness and high sensitivity of LAMP-assisted snail diagnosis was later confirmed in Brazil by Gandasegui et al. [16] and in China by Kumagai et al. [17]. The latter research group developed a diagnostic platform based on a target 28S ribosomal DNA (rDNA) specific for S. japonicum (not reacting with S. mansoni) and showed that snails experimentally infected with only one miracidium could be detected less than 24 hours after infection.'

Point 7: Line 81  Replace with "studied".

Response 7:

Thanks, we changed 'described here' to 'studied'.

Point 8: Line 81  Can you add a line or two about the pooled snail DNA samples to make it clear for the readers

Response 7:

Changed the sentence to 'We studied the application of LAMP using pooled snail samples, i.e. instead of testing each snail separately, we combined snails, however never in numbers exceeding 50 per pooled sample based on preliminary dilution tests (see Fig.  2).' starting the next sentence as follows: 'The protocol used was derived from a of visual LAMP detection method developed by Tomita et al. [18], where the amplification of the pyrophosphate ion by-product combines with a divalent metal ion to form an insoluble salt.'

Point 8: Line 83  Insert "of" before pyrophosphate.

Response 8:

Thanks, we insert "of" before pyrophosphate.

Point 9: Line 86-88

This section needs change to correct the tense. Also, at the end "arbiter" is not the correct word choice for DNA sequencing, as it is only determining the id of the amplified product, not resolving False positive vs. True Positive.

Response 9:

We changed to 'We compared the results of the LAMP assay with the outcome using the polymerase chain reaction (PCR) applying DNA sequencing to determine the identity of the amplified product.

Point 10: Line 94 add "by microscopy" at the end.

Response 10:

We insert 'LAMP' before 'procedure', I think you might misunderstood, so I didn't add microscopy at the end.

Materials and Methods

Point 11: Line 97-99 This sentence should be in the past tense, as this is already been done.

Response 11:

Thanks, We changed to 'This study constitutes an evaluation of LAMP-based snail diagnosis,eventually aimed to be part of a platform integrating different kinds of data enabling improved surveillance of schistosomiasis transmission.'

Point 12: Line 167-168 The meaning is not clear for this section.

Response 12:

Yes, We described in more detail. Changed to “observing the tubes with the unaided eye because the color of the reaction changed from orange to yellow-green in the presence of LAMP amplicon (Figure 2,3)”.

Point 13: Line 170 Replace with life cycle stages of S. japonicum

Response 13:

Thanks, we replaced 'parasite stages' with life cycle stages of S. japonicum.

Point 14: Line 177 Correct the font and explain why this one is used for specificity.

Response 14:

 Yes, we described what was done in detail and explained why this one is used for specificity.

Point 15: Line 179 Delete "s".

Response 15:

Thanks, deleted “s”.

Point 16: Line 180 Delete Capacity

Response 16:

Yes, deleted “Capacity”.

Point 17: Line 187 Delete. a

Response 17:

Yes, deleted “a”.

Point 18: Line 189 Delete respectively

Response 18:

Yes, deleted respectively.

Point 19: Line 193-203 Are these samples validated against actual result kept in NIPD, as mentioned all the laboratory personnels are blinded. In that case please mention that in the text.

Response 19:

Correct! these samples validated against actual result kept in NIPD (we call “reference results”). Also other details we mentioned in the text.

3. Results

Point 20: Line 223-227  This section needs revision. The meaning s not clear.

Response 20:

Thanks, We changed to “The results further demonstrated that the schistosome DNA included in a single miracidium is sufficient to be amplified by the LAMP assay making it possible to detect the schistosome infection already at the sporocyst stages in the snail as well when mature cercariae are ready for release. The Trichobilharzia cercariae used to test the specificity were negative. (Figure 3)

Point 21: Figure 3

If possible please add positive and negative on the picture for better understanding for the reader.

Response 21:

Thanks. We added positive and negative on the update Figure 3.

Point 22: Table 2  Explain the acronym Pos.

Response 22:

Thanks. We did it the updated table 1 (formerly Table 2).

3. Discussion

Point 23: Line 267-274

This section needs correction and rearrangement of sentences to make the clear meaning of this segment. There are some mistakes too, such as on Line 271-272: snail infected snail.

Response 23:

Changed to On the other hand, traditional snail diagnosis depends on labour-intensive, time-consuming individual dissection of thousands of collected snails from the field [19], and the results are not sufficiently sensitive to support this statement.  However, the advent of molecular diagnostics changed all that. The successful use of the LAMP technique shown here promises to revolutionize snail surveillance, not only in the laboratory but also in field [13-17]. The reported results are timely as the use of snail control as a complementary approach along with chemotherapy is now again brought forward as a perhaps necessary means to achieve elimination of schistosomiasis [20].

Point 24: Line 285-288  Please also discuss that LAMP can be evaluated with simple SYBR Green addition with 1/20 or 1/30 dilution, which can also be kept in room temperature without requirement of refrigeration. This might be even cheaper than calcein method. As resource and financial limitation is the focus of this manuscript, it is important to discuss other alternatives.

Response 24:

Changed toThe development of a surveillance platform based on molecular diagnostic techniques characterized by simplicity and reliability, yet high throughput, requires an approach adaptable to county-level laboratories with limited resources. The manganese/calcein method of Tomita et al. [18] is an important contribution in this regard as it enables recognition of small quantities of DNA by means of a fluorescent signal emitted from the sample solution after amplification that was successfully used in China by Kumagai et al. [17]. Other attempts to increase readability and sensitivity include the use of various dyes, e.g., SYBR green (Singh et al., 2017) or hydroxyl napthol blue (Ali et al., 2017) have has also produced good results. Dyes have the advantage of being independent of refrigeration. Importantly, we are already in a situation where more investigation might provide the needed room for improvement. Any increase of the dilution factor would accelerate the areal needed to be tested considerably [24]. We found a dilution ratio of 1/50 to be useful (Figure 2), but if the number of infected snails decreases, as it is supposed to do with the elimination programme in force, the risk of mistakenly declare an area free of transmission increases if the snail pool contains too few snails. This could be counteracted by using a higher number of snails in the pool and increasing the time of amplification for the test but, on the other hand, the risk for false positives would then rise.

Point 25: Line 289-292 This section is confusing. Please rewrite this sentence.

Response 25:

Yes, have already changed, as Response 24.

Point 26: Line 293-297  This is a complex long sentence. Please break it down into small sentences to make meaning clear to the readers.

Response 26:

Yes, have already changed, as Response 24.

Point 27: Line 298 "m" instead of M.

Response 27:

Yes, changed "m" to M.

Point 28: Line 300   Indentation is not right.

Response 28:

Sorry, I didn’t understand the meaning.

Point 29: Line 305-308  Rewrite this sentence.

Response 29:

Change to 'A further advantage of the LAMP technique, although highly technical, is easy to perform in basic laboratory settings common in the rural areas. It is also easy to learn as shown by the excellent agreement over three years between the many places included in this study. In addition, while sporocysts and germ cell balls.....'

Point 30: Line 308  Is it "balls" or "cells"?

Response 30:

Cell balls.

Point 31: Line 311  Replace with "is"

Response 31:

Yes, replaced.

Point 32: Line 31There should some discussion about the shortcomings of the microscopy and highlighting of the outcome of the study, as this is the main focus.

Response 32:

Changed to 'As seen in Table 1, the outcome of testing with PCR and the LAMP approach agreed largely, which means that the sensitivity of the two techniques is the same in practice However, theoretically, the latter has the edge since the one sample reacting negatively by PCR was positive when tested with the LAMP test. Comparison with microscopy, on the other hand, clearly showed that molecular testing is superior and should be use in the future. The excellent agreement between all 28 field laboratories (Table 2) shows we are now ready to change from microscopy to molecular testing, preferably using LAMP as it lends itself to use in the field.  It is suggested that this kind of snail testing be included together with diagnostic testing of humans and domestic animals in a joint surveillance and response platform based on only high-resolution techniques.

4. Conclusion

Point 33: Line 325  Replace with "endemic".

Response 33: Thanks. Replaced.

Point 34: Line 325 Is there any plan to address these issues in future?

Response 34: Yes. We are trying to optimize the LAMP reagent (i.e. lyophilization) in order to make the reagent more easy transportation and stable. We are doing it and will use the lyophilized reagent in the field next year.

Reviewer 2 Report

The manuscript “Field evaluation of a loop-mediated isothermal amplification (LAMP) platform for the detection of Schistosoma japonicum infection in Oncomelania hupemsis snails” evaluates the field-applicability of a LAMP-based detection method for S. japonicum, and compares the method sensitivity and accuracy with microscopy and PCR. The manuscript provides a validated detection tool for schistosomiasis with a much higher sensitivity than traditional microscopy. This is an important next step in improving snail monitoring methods for detection of possible risk of transmission. This is also in line with control efforts shifting towards environmental control from years of drug-based treatment in humans.

The study is overall well-structured with an impressive field-work, and results are well analyzed and discussed. However, the presentation and description of methods and results needs clarification and more details. Additionally, the English language requires moderate revision, possibly using a native English speaker. Therefore, I suggest it should be published in Tropical Medicine and Infectious Disease after major revisions and addressing the following comments:

Abstract:

Line 22-23: The sentence “The results were compared with those……” should be rewritten since it is unclear what is meant.

Line 27-28: How are these two infection rates, 6.5% (LAMP) and 0,04% (microscopy), calculated? They only appear in the abstract, and is found nowhere in the methods or results section. Please put in details on this in the main text as well. Are they calculated from the results in table 2?

Introduction:

Line 70-80: This sentence is 10 lines long, and it is not very clear. Please divide into shorter sentences and rephrase for easier understanding.

Line 76: The name of the first author of ref number 17 is misspelled. The author name is Kumagai, and not Kumagagi. Please correct throughout main text and in the reference list.

Line 87-88: The meaning of this sentence is unclear, please rewrite.

Line 89-94: The aim is clear and concise.

Materials and methods:

Line 98: Enabling what translational research? Not clear what is meant.

Line 102: It is very helpful with a flow chart for LAMP and PCR (Figure 1). But unfortunately the figure is not clear for the lower part of the figure, after ‘Result visualization’. I get that the result is either negative (-) or positive (+) based on color change. But how does the ‘PCR and DNA sequence’ fit into the flow chart? Why is the arrow pointing upwards? When is the PCR carried out and on what samples? Please revise the figure 1 for easier understanding of the approach.

Line 111: There are no details or descriptions on how snails were collected? Using scooping for 20 min or what? Which type of water bodies was sampled – streams, ponds? Please describe or put in a reference describing the snail sampling method.

Line 112-114: Maybe a map would be useful here as the authors have performed a very substantial field work. It is just a suggestion.

Line 114-119: Long sentence, please revise and correct the English language. Maybe put in a reference for the microscopy method.

Line 119-120: Please correct the language.

Line 123: It says that the tissue collections were homogenized. How was that done? Please explain.

Line 126: ….then stored at -20C and until use……Please delete the word ‘and’ in that sentence.

Line 129-141: It is not clear where PCR was performed on exactly the same samples as the ones analyzed with LAMP (this is not clear from Figure 1 either). Please clarify here in the main text and in the figure 1, as mentioned earlier.

Line 132-133. There is no reference for the primers, which usually means that the authors designed them themselves. But it seems that the primers are designed by Kumagai et al 2010, so please put in reference.

Line 134: Please put in primer concentration used. Only the volume is described.

Line 143-155: Please state how many samples were sequenced. Was it all of them or how many percent?

Line 153-155: The last part of the sentence after ‘….search tool (BLAST)’ should be deleted. No need to explain what BLAST does.

Line 163-165: Please rephrase sentence due to language.

Line 172-173: Meaning not clear, please rephrase.

Line 174-177: The authors write that they carried out infection experiments with the intermediate snail host. How was that done? Please describe what was done? Was is lab-based snails or filed-collected? How many miracidia per snail and where did the miracidiae come from? How was the snails kept and what was the prepatent period?

Line 177: The last sentence is written in smaller font size. Please correct.

Line 183: The schistosome-positive/negative snail ratios are not the same as presented in the legend for Figure 2. Please correct.

Line 185: It is unclear what is meant by ‘(though separate from one and another)’? Please correct language.

Line 184-190: As mentioned before, the snail infection rates of 6.5% (LAMP) and 0,04% (microscopy) should be described in the main text. But how can the infection rate be calculated if all the snails were pooled before DNA extraction? Please explain.

Results:

Line 206-208: Please refer to Figure 2 in the main text.

Line 219-220: Why do the authors refer to Table 2 as the first table. Please order the tables correctly and the table references in the main text accordingly. Maybe table 2 is supposed to be table 1?

Line 217-223: This text presents somewhat the same results as the text in line 249-255 as they both refer to table 2. Please structure the results section properly to avoid repeating text.

Line 220-223: Please rephrase this sentence – too many information in one sentence with unclear language. The meaning is not clear, especially since the total numbers of negative and positive samples are not found in table 2. Consider putting in total numbers in the table for easier understanding. For example, the text says ’14 positve samples for both assays’ – please write LAMP and PCR assays. But there were 15 positive samples for LAMP according to table 2? And what do the authors mean by ‘the remaining single sample was negative by PCR but positive by DNA sequence analysis’? – was all the negative PCR samples also sequenced?

Line 227: The last sentence is written in smaller font size. Please correct.

Line 234-238: Long sentence, so please divide into shorter for easier understanding.

Line 238: Please correct 2011 to 2014. And please correct the table reference – should it be 1 or 2? And in table 1, the last CDC lab from Chongqin Province don’t have any score results, so please consider removing from the table.

Line 238-241: The authors state that labs with lower scores than 100 inone year did not have that in other years. From the table it is evident that Dali lab from Yunnan Province had the scores 80 in 2014 and 90 in 2015. So please revise text accordingly.

Line 254-255: The single snail pool that was positive by LAMP and negative by PCR – was that the same PCR that was sequenced and proved to be positive? The sample mentioned in line 221-222?

Discussion:

Line 283-285. Please rephrase due to unclear language.

Line 286: The Tomita et al ref – is that number 17 or 18? Earlier it is referred to as number 18.

Line 289: The Kumagai et al ref – is that number 17 or 18? Earlier it is referred to as number 17. Please go through reference list and correct numbers and author names.

Line 290: if=of

Line 298: More should be in capital.

Line 312-313. Please rephrase, meaning of sentence unclear.

Consider discussing how this LAMP method would fit into already exisitng control efforts and strategies. Are snail monitoring a part of Chinas control efforts? How much and how often would snail Collections be needed to monitor risk of transmission? Is that feasable in the present setting?

Author Response

Response to Reviewer 2 Comments

Dear Professor,

We appreciate you a lot for your nice comments.

We did a thorough revision of the manuscript, and addressed your comments point by point as followings.

Thanks and regards,

Zhiqiang Qin

Abstract:

Point 1: Line 22-23: The sentence “The results were compared with those……” should be rewritten since it is unclear what is meant.

Response 1: Thanks. We added “ in addition” before the sentence. We think it can make senese.

Point 2: Line 27-28: How are these two infection rates, 6.5% (LAMP) and 0,04% (microscopy), calculated? They only appear in the abstract, and is found nowhere in the methods or results section. Please put in details on this in the main text as well. Are they calculated from the results in table 2?

Response 2: Thanks. Yes, we calculated 6.5% (15/232), 0.04% (2/232), respectively. In detail as displayed in Table 1 (formerly Table 2).

 Introduction:

Point 3: Line 70-80: This sentence is 10 lines long, and it is not very clear. Please divide into shorter sentences and rephrase for easier understanding.

Response 3: Change to 'The loop-mediated isothermal amplification (LAMP) technology employs a polymerase that amplifies the target DNA gene sequence with high specificity and rapidity under isothermal conditions [14]. Hamburger and colleagues investigated the use of this technique for the detection of infections due to S. mansoni and S. haematobium showing excellent results, not only confirming that the LAMP technique works in the laboratory, but also in the field in Africa [15]. The usefulness and high sensitivity of LAMP-assisted snail diagnosis was later confirmed in Brazil by Gandasegui et al. [16] and in China by Kumagai et al. [17]. The latter research group developed a diagnostic platform based on a target 28S ribosomal DNA (rDNA) specific for S. japonicum (not reacting with S. mansoni) and showed that snails experimentally infected with only one miracidium could be detected less than 24 hours after infection.'

Point 4: Line 76: The name of the first author of ref number 17 is misspelled. The author name is Kumagai, and not Kumagagi. Please correct throughout main text and in the reference list.

Response 4: Thanks! We changed to the correct name “Kumagai” in the main text and in the reference No.16.

Point 5: Line 87-88: The meaning of this sentence is unclear, please rewrite.

Response 5: Changed to 'We compared the results of the LAMP assay with the outcome using the polymerase chain reaction (PCR) applying DNA sequencing to determine the identity of the amplified product.

Point 6: Line 89-94: The aim is clear and concise.

Response 6:  Thanks a lot. The study aims to three objectives: Firstly, it is for testing the pooled samples is better for single one testing? Secondly, it is to confirm the sensitivity of LAMP method in detecting Schistosoma DNA in snail samples in known endemic area. Thirdly, it is to investigate and validate its application under field conditions soon after snail collection.

 Materials and methods:

Point 7: Line 98: Enabling what translational research? Not clear what is meant.

Response 7: Yes, Changed to 'This study constitutes an evaluation of LAMP-based snail diagnosis,eventually aimed to be part of a platform integrating different kinds of data enabling improved surveillance of schistosomiasis transmission.'

Point 8: Line 102: It is very helpful with a flow chart for LAMP and PCR (Figure 1). But unfortunately the figure is not clear for the lower part of the figure, after ‘Result visualization’. I get that the result is either negative (-) or positive (+) based on color change. But how does the ‘PCR and DNA sequence’ fit into the flow chart? Why is the arrow pointing upwards? When is the PCR carried out and on what samples? Please revise the figure 1 for easier understanding of the approach.

Response 8: I appreciate for the comment. For organize Figure 1, I tried to make the figure better understood by removing the frame at the bottom 1) quality control and identification and added the frame of PCR and DNA sequence in the direct way.

Point 9: Line 111: There are no details or descriptions on how snails were collected? Using scooping for 20 min or what? Which type of water bodies was sampled – streams, ponds? Please describe or put in a reference describing the snail sampling method.

Response 9: Sorry for that. We added a reference 19 in the text. The snail sampling methods were described clearly in reference 19, somehow resemble a standard operation manual in China.

Point 10: Line 112-114: Maybe a map would be useful here as the authors have performed a very substantial field work. It is just a suggestion.

Response 10:  We agree with the suggestion that it would be useful if it added a map into the text. However, the main objective of this paper is to explore the sensitivity and practicable of LAMP method in detecting the Schistosoma DNA in the pooled snail samples. Sorry for that we still not make a map here.

Point 11: Line 114-119: Long sentence, please revise and correct the English language. Maybe put in a reference for the microscopy method.

Response 11: Thanks. Changed to 'After having been crushed by pressure between clean glass plates, and the pieces of shell removed, the snails were examined, one by one, under the microscope at low magnification (generally 10X) to certify whether cercariae and/or sporocysts were present. The snail bodies were afterwards used for testing by the LAMP assay and conventional PCR'.

Point 12: Line 119-120: Please correct the language.

Response 12: Yes, we double checked.

Point 13: Line 123: It says that the tissue collections were homogenized. How was that done? Please explain.

Response 13: Thanks. We added Briefly, the snail bodies were pooled in clean 2.0 or 10.0 ml tubes (not exceeding 50 snails in each),adding 400-1000 μL of DNA lysis buffer (Qiagen, Valencia, CA, USA ) into each tube and the tissue collections homogenized,  20-50 μL of ProteinK (Qiagen, Valencia, CA, USA)was then added to the snail homogenates and the tubes kept in a water bath at 56°C, after 2-3 hours of incubation, the supernatant of snail homogenates were saved and passed through from the DNA binding column (Qiagen, Valencia, CA, USA), the dedicated DNA binding column was washed by the wash buffer (Qiagen, Valencia, CA, USA) after centrifuged, the genomic DNA was recovered through eluted the DNA binding column by using nuclease free water. The genomic DNA then stored at -20 °C until use. 

Point 14: Line 126: ….then stored at -20C and until use…Please delete the word ‘and’ in that sentence.

Response 14: Yes, we deleted the word “and”.

Point 15: Line 129-141: It is not clear where PCR was performed on exactly the same samples as the ones analyzed with LAMP (this is not clear from Figure 1 either). Please clarify here in the main text and in the figure 1, as mentioned earlier.

Response 15: Yes, we pointed out wheterg PCR was performed on exactly the same samples, Figure 1.

Point 16: Line 132-133. There is no reference for the primers, which usually means that the authors designed them themselves. But it seems that the primers are designed by Kumagai et al 2010, so please put in reference.

Response 16: Thanks! Yes, the primers are designed by Kumagai et al (2010) and we put the reference 17 in the text.

Point 17: Line 134: Please put in primer concentration used. Only the volume is described.

Response 17: Yes, we put primer concentration (10 pmol/L) used in the text.

Point 18: Line 143-155: Please state how many samples were sequenced. Was it all of them or how many percent?

Response 18: Thanks. Totally of 15 positive samples were send to DNA sequence. 14 for both PCR and LAMP testing positive, and only one is LAMP reaction positive while PCR negative reaction.

Point 19: Line 153-155: The last part of the sentence after ‘….search tool (BLAST)’ should be deleted. No need to explain what BLAST does.

Response 19:Full stop after 'database' 'delete 'using the basic local alignment search tool (BLAST).....matches’

Point 20: Line 163-165: Please rephrase sentence due to language.

Response 20: Changed to 'In addition, 23 μL of the reaction mixture was added to the Sj28S rDNA target in one tube (the positive control) and also to nuclease-free water in another tube (the negative control)'.

Point 21: Line 172-173: Meaning not clear, please rephrase.

Response 21: Changed to 'To confirm whether DNA from a single miracidium could be detected by the LAMP assay, we used five different positive samples and performed the test with DNA collected from one miracidium from each of them using the extraction and purification kit.

Point 22: Line 174-177: The authors write that they carried out infection experiments with the intermediate snail host. How was that done? Please describe what was done? Was is lab-based snails or filed-collected? How many miracidia per snail and where did the miracidiae come from? How was the snails kept and what was the prepatent period?

Response 22: Yes, we described what was done in detail.

Point 23: Line 177: The last sentence is written in smaller font size. Please correct.

Response 23: Yes, corrected font and explain why this one is used for specificity.

Point 24: Line 183: The schistosome-positive/negative snail ratios are not the same as presented in the legend for Figure 2. Please correct.

Response 24: The schistosome-positive/negative snail ratios are not the same as presented in the legend for Figure 2. Yes, we have changed the schistosoma-positive/negative snail ratios to the same as presented in the legend for Figure 2.

Point 25: Line 185: It is unclear what is meant by ‘(though separate from one and another)’? Please correct language.

Response 25: Changed to  ‘......(though the different pools investigated contained different sets of snails)’

Point 26: Line 184-190: As mentioned before, the snail infection rates of 6.5% (LAMP) and 0,04% (microscopy) should be described in the main text. But how can the infection rate be calculated if all the snails were pooled before DNA extraction? Please explain.

Response 26: Before DNA extraction, we put the snails in the clean glass plate (the snails from the same environment of endemic area), and detected if or not the parasite inside the pooled snails' body using microscopy. And the pooled size of fifty, except no enough numbers of snails in the same environment. After microscopy screening, we collected the tissue of pooled snails for further DNA extraction and as well as LAMP detection. In addition, the denominator should be the same for both microscopy testing and LAMP, PCR testing.

 Results:

Point 27: Line 206-208: Please refer to Figure 2 in the main text.

Response 27: Yes, Add '(Fig. 2)' after 'results'.

Point 28: Line 219-220: Why do the authors refer to Table 2 as the first table. Please order the tables correctly and the table references in the main text accordingly. Maybe table 2 is supposed to be table 1?

Response 28: Thanks! We changed to Table 1 here and shift the tables on the page so that Table 2 becomes Table 1 and vice versa. We did it as follows:

The text on lines 217-222 be changed to:

'The ready-to-use LAMP test kit evaluated  here  detected 7.5 times more infected snails than did microscopy, while the PCR results were in agreement with those of the LAMP assay, except with respect to one single snail pool that was positive by LAMP  and negative by PCR.  Out of 232 pooled samples tested, 217 were negative and 14 positive by both assays (Table 1), while the remaining single sample positive by LAMP but negative by PCR was indeed  positive by DNA sequence analysis, underlining LAMP's superiority.'to be followed by Table 1 (i.e. the former Table 2) to which a bottom line showing the totals should be added.

Then  the current lines 223-227 should come: 'The results further demonstrated that the schistosome DNA included in a single miracidium is sufficient to be amplified by the LAMP assay making it possible to detect the schistosome infection already at the sporocyst stages in the snail as well when mature cercariae are ready for release (Figure 3). The Trichobilharzia cercariae used to test the specificity were negative (correct font for this sentence)' would follow with Fig. 3 that belongs together with this paragraph.

Point 29: Line 217-223: This text presents somewhat the same results as the text in line 249-255 as they both refer to table 2. Please structure the results section properly to avoid repeating text.

Point 30: Line 220-223: Please rephrase this sentence – too many information in one sentence with unclear language. The meaning is not clear, especially since the total numbers of negative and positive samples are not found in table 2. Consider putting in total numbers in the table for easier understanding. For example, the text says ’14 positve samples for both assays’ – please write LAMP and PCR assays. But there were 15 positive samples for LAMP according to table 2? And what do the authors mean by ‘the remaining single sample was negative by PCR but positive by DNA sequence analysis’? – was all the negative PCR samples also sequenced?

Point 31: Line 227: The last sentence is written in smaller font size. Please correct.

Point 32: Line 234-238: Long sentence, so please divide into shorter for easier understanding.

Point 33: Line 238: Please correct 2011 to 2014. And please correct the table reference – should it be 1 or 2? And in table 1, the last CDC lab from Chongqin Province don’t have any score results, so please consider removing from the table.

Point 34: Line 238-241: The authors state that labs with lower scores than 100 inone year did not have that in other years. From the table it is evident that Dali lab from Yunnan Province had the scores 80 in 2014 and 90 in 2015. So please revise text accordingly.

Response 29-34: Then comes the following text including Table 2 (formerly Table 1):

Lines 234-240 - Changed to 'The 28 different laboratories at the provincial and county health agency levels belonging to the inter-laboratory panel demonstrated an almost total agreement with respect to the results over the three year reported. Only four laboratories showed slightly lower than the others' 100 score, one with score 80 in 2013, two with score 60 in 2014 and one with score 80 in 2014 and score 90 in 2015 (Table 2). Importantly, with the exception of one laboratory, those with lower scores than 100 in one year, did not have that in other years. The number of samples in 2013 was not sufficient to be tested in all laboratories as signified with a dash in the table.'  In addition, I removed the last laboratory (Chongqin) as it did not participate in any of the years. And I widen column 2 in the table so 'Laboratory' can fit.

The current lines 249-255 can all be deleted as the contents have been baked into lines 217-222 above and Table 1 (i.e. the former Table 2) moved upward.

Here are the three tables with some suggested changes (numbering of the tables and added text).  Especially, a negative control results were added in the form of Table 3.

Point 35: Line 254-255: The single snail pool that was positive by LAMP and negative by PCR – was that the same PCR that was sequenced and proved to be positive? The sample mentioned in line 221-222?

Response 35: Thanks! The single snail pool that was positive by LAMP and negative by PCR, we run the Agarose gel analysis of LAMP reaction product and cut the band to DNA sequence. But the other 14 positive samples against PCR and LAMP reaction were sent for DNA sequence by using the recovered DNA after PCR reaction.

 Discussion:

Point 36: Line 283-285. Please rephrase due to unclear language.

Point 37: Line 286: The Tomita et al ref – is that number 17 or 18? Earlier it is referred to as number 18.

Point 38: Line 289: The Kumagai et al ref – is that number 17 or 18? Earlier it is referred to as number 17. Please go through reference list and correct numbers and author names.

Point 39: Line 290: if=of

Point 40: Line 298: More should be in capital.

Response 36-40:

Lines 283-299 - Changed toThe development of a surveillance platform based on molecular diagnostic techniques characterized by simplicity and reliability, yet high throughput, requires an approach adaptable to county-level laboratories with limited resources. The manganese/calcein method of Tomita et al. [18] is an important contribution in this regard as it enables recognition of small quantities of DNA by means of a fluorescent signal emitted from the sample solution after amplification that was successfully used in China by Kumagai et al. [17]. Other attempts to increase readability and sensitivity include the use of various dyes, e.g., SYBR green (Singh et al., 2017) or hydroxyl napthol blue (Ali et al., 2017) have has also produced good results. Dyes have the advantage of being independent of refrigeration. Importantly, we are already in a situation where more investigation might provide the needed room for improvement. Any increase of the dilution factor would accelerate the areal needed to be tested considerably [24]. We found a dilution ratio of 1/50 to be useful (Figure 2), but if the number of infected snails decreases, as it is supposed to do with the elimination programme in force, the risk of mistakenly declare an area free of transmission increases if the snail pool contains too few snails. This could be counteracted by using a higher number of snails in the pool and increasing the time of amplification for the test but, on the other hand, the risk for false positives would then rise.

Point 41: Line 312-313. Please rephrase, meaning of sentence unclear.

Consider discussing how this LAMP method would fit into already exisitng control efforts and strategies. Are snail monitoring a part of Chinas control efforts? How much and how often would snail Collections be needed to monitor risk of transmission? Is that feasable in the present setting?

Response 41:

Changed to 'As seen in Table 1, the outcome of testing with PCR and the LAMP approach agreed largely, which means that the sensitivity of the two techniques is the same in practice However, theoretically, the latter has the edge since the one sample reacting negatively by PCR was positive when tested with the LAMP test. Comparison with microscopy, on the other hand, clearly showed that molecular testing is superior and should be use in the future. The excellent agreement between all 28 field laboratories (Table 2) shows we are now ready to change from microscopy to molecular testing, preferably using LAMP as it lends itself to use in the field.  It is suggested that this kind of snail testing be included together with diagnostic testing of humans and domestic animals in a joint surveillance and response platform based on only high-resolution techniques.

Reviewer 3 Report

This MS is a test of quality for a LAMP to Schistosoma japonicum, using snails collected in the field of endemic areas in China.

The experimental design is somehow complicated to understand in terms of the reasons for the experiments done in this study.

One or more groups of snails from a non-endemic area could be tested for confirmation on the specificity of the used test. 

The kits and other materials names in detail are missing difficulting the reproducibility of this work. 

There are no references for the primers used in the PCR, and the description of the method doesn't give to the reader exact concentrations used (sometimes states only the used volume) 

LAMP assay also has no details of the materials and a careful description of the methodology.

How the authors reached the values of 1:50 infected snail as a minimum infection rate is if 1)the positive snails were selected by microscopy, however, microscopy, as the authors stated has low sensitivity. Some infected snails not diagnosed by microscopy could be in the pooled samples jeopardizing the calculation. 

In the results, the authors stated negative PCR were sequenced and confirmed to be S. japonicum. How samples could be sequenced from negative PCRs?

The specificity control was described to be done using Trichobilharzia cercarie, a more detailed description of DNA concentration is necessary in this case.

Some references are cited wrong in the text.

Author Response

Response to Reviewer 3 Comments

Dear Professor,

We appreciate you a lot for your nice comments.

We did a thorough revision of the manuscript, and addressed your comments point by point as followings.

Thanks and regards,

Zhiqiang Qin

Point 1: This MS is a test of quality for a LAMP to Schistosoma japonicum, using snails collected in the field of endemic areas in China.

Response 1:  Thanks. Yes, we detected the S.japonicum DNA in the snail samples in which collected the field of endemic areas of Hunan, Hubei, Jiangxi, Anhu and as well as Yunnan five provinces in China. In addition, we also tested the snail samples that collected from non-endemic areas of Zhejiang, Shanghai and Guangxi provinces in China. Taken together, we using snails collected in the field of endemic areas and non-endemic area (before as endemic area, but eliminated last for almost 20 years) in China.

Point 2: The experimental design is somehow complicated to understand in terms of the reasons for the experiments done in this study.

Response 2: Sorry for that. Basically, the main aim of this study is to investigate and validate its application for LAMP method under field conditions soon after snail collection. Basing on that, firstly, we tried to confirm the sensitivity of LAMP method in detecting Schistosoma DNA in snail samples in known endemic area, Secondly, for quality control of this new method, we did the conventional method microscopy and PCR against the same samples at the same time. Finally, all of the positive reaction samples against PCR and/or LAMP will be identified by DNA sequence analysis. In addition, the study also aimed to testing the pooled samples is better for single one testing.

Point 3: The kits and other materials names in detail are missing difficulting the reproducibility of this work. There are no references for the primers used in the PCR, and the description of the method doesn't give to the reader exact concentrations used (sometimes states only the used volume) 

Response 3: We agree with your comments. We re-write in detail in 2.5 LAMP assay the text so can provide the reproducibility of this work, as described in Response 4. We also put the concentration of PCR primers in the text and a reference number in the text.

Point 4: LAMP assay also has no details of the materials and a careful description of the methodology.

 Response 4: The descriptions contains all details normally given for this type of assays. However, we agree with your comment, as we put further details in the 2.5.4 LAMP assay as followings,

  (A) Reagent Setup and Primer sequence

a. Reaction mixture (2× composition): Tris buffer (pH 8.8) 40 mM,,KCL (20 M), MgSO4 (16 mM), (NH4)2SO4 20 mM, Tween 20 0.2%),Betaine 1.6 M),dNTPs 2.8 mM each.

b. Calcein working soulution (2× composition):   Calcein (50 Um), MnCl2(1mM).

c.Sj28S gene Primers [17] (5'-3';Sangon; HPLC purification)    F3(GCTTTGTCCTTCGGGCATTA), B3(GGTTTCGTAACGCCCAATGA), FIP (ACGCAACTGCCAACGTGACATACTGGTCGGCTTGTTACTAGC),BIP(TGGTAGACGATCCACCTGACCCCTCGCGCACATGTTAAACTC)

(B) Procedure-Amplification (60-90 min)

We performed  a reaction mixture of 7.5 μL nuclease-free water, 2×12.5 μL reaction buffer, 1.0 μL primer mixture (25×), 1.0 μL (8 μ/μL) Bst, a Bacillus stearothermophilus DNA polymerase homologue used for DNA strand displacement, and 1.0 μL calcein. We used 0.2 ml reaction tubes dispensing 23 μL of the reaction mixture together with 2.0 μL of the sample to be tested into each tube. In addition, 23 μL of the reaction mixture was added to the Sj28S rDNA target in one tube (the positive control) and also to nuclease-free water in another tube (the negative control). All reaction tubes were incubated at 65°C for 6090 min, followed by inactivation of the enzyme by incubation of the tubes at 85°C  for 5 min and then observing the tubes with the unaided eye because the colour of the reaction changed from orange to yellow-green in the presence of LAMP amplicon (Figures 2,3).

Point 5: How the authors reached the values of 1:50 infected snail as a minimum infection rate is if 1) the positive snails were selected by microscopy, however, microscopy, as the authors stated has low sensitivity. Some infected snails not diagnosed by microscopy could be in the pooled samples jeopardizing the calculation. 

Response 5:  We agree with comment.  Well, while we designed this experiment, we used the lab-bred snails and we already make sure which one is Schistosoma infected and which one is negative. The protocol as followings:

Firstly of all, we collected snails in the filed from non-endemic areas (schistosomiasis japonica elimination) in China. All of the snails were keeping culture in laboratory for one month. Then, we put the snails into the Chlorine free water to detect if or not Schistosoma cercariae inside the snail body. Yes, all of the bred snails were totally Schistosoma negative. Secondly, we challenge infected the lab-bred snails (parts of) with Schistosoma japonicum miracidia. After challenge infection, we then identified the mother sprocyst, daughter sprocyst as well as cercariae in different bred time. Finally, we will pick up the Schistosoma infected snails and negative snails for further pooled sampling study in Lab. Therefore, we are believe that we reached the values of 1:50 infected snail as a minimum infection rate should be meet the requirements for next further field investigation.

Point 6: In the results, the authors stated negative PCR were sequenced and confirmed to be S. japonicum. How samples could be sequenced from negative PCRs?

Response 6: Thanks!  One LAMP positive reaction sample but negative PCR. In this case, we run the agarose gel analysis for the LAMP reaction products, and then to isolate the target band. In the end, the purified product of the recover target band was send for DNA sequence analysis.

Point 7: The specificity control was described to be done using Trichobilharzia cercarie, a more detailed description of DNA concentration is necessary in this case.

Response 7: In this case, we used the same methodology as described above under 2.3 and 2.4 and shown in Figure 1. A sentence to that effect has been added.

Point 8: Some references are cited wrong in the text.

Response 8: Thanks. We checked all the cited reference and corrected it.

Round 2

Reviewer 1 Report

I am pleased to see the improvement of the manuscript and now it is a well organized manuscript that will surely interest readers. The authors did a good job addressing all the major and minor issues being raised. Please do a final check in terms of some composition, grammar and parenthesis.

Author Response

Response to Reviewer 1 Comments

Dear Professor,

We appreciate you a lot for your nice comments.

We did a thorough revision of the manuscript, and addressed your comments point by point as followings.

Thanks and regards,

Zhiqiang Qin

Point: I am pleased to see the improvement of the manuscript and now it is a well organized manuscript that will surely interest readers. The authors did a good job addressing all the major and minor issues being raised. Please do a final check in terms of some composition, grammar and parenthesis. 

Response: We appreciate a lot!  We did a final check through the whole manuscript according to your comments. There are some items were corrected as followings:

One is that we found that the description of “snail sampling method” it is not very clear what was done. Now we have rephrased it and put in 2 references (No.20 and No.21) for the sampling method.

Another is that we have asked the native English Speaking expert to help us to correct English language and as well as grammar issues before we submitted this time.

Reviewer 2 Report

Reviewer response 2 for manuscript tropicalmed-383709

 I am happy to see that the English language has been improved. Only a few corrections remain. The flow in the main text is good and it is easy to understand. The revised Figure 1 is good. Therefore, I suggest it should be published in Tropical Medicine and Infectious Disease after minor revisions and addressing the following comments:

Line 119-124: It is good that a section on the snail sampling procedure has been put in the main text, however it is not very clear what was done. It is confusing with the distance measurements – maybe too much detail?! Please rephrase, put in 1-2 references for the sampling method, and please correct English language, grammar issues (capital letters after comma etc).

Line 131: put in spacing around the bracket “(tissues)”

Line 117-120: Ad response 9: There is only a ref 18 in the section on snail sampling. And that ref is put in to support the statement on the infection rates. If it is the same ref, then put it in again when you describe the snail sampling, e.g. in line 120.

Line 143: put space in after 3’

Line 159-162: Please rephrase or correct the English language – not clear.

Line 183: The snail species name has been written earlier so write O. hupensis. Please check throughout main text.

Table 1: For the Pos. Microscopy results, it shows that there is 1 for Hubei province, but the total shows 2 – how is that possible?? Please correct results – this is very important.

Line 247 and table 3: It says 28 different laboratories, but there is only 27 mentioned in the table. I guess you removed the last CDC lab from Chongqin Province from the table 2, but please revise text accordingly throughout the manuscript. Is it 27 or 28 labs?

Author Response

Response to Reviewer 2 Comments

Dear Professor,

We appreciate you a lot for your nice comments.

We did a thorough revision of the manuscript, and addressed your comments point by point as followings.

Thanks and regards,

Zhiqiang Qin

Point 1: Line 119-124: It is good that a section on the snail sampling procedure has been put in the main text, however it is not very clear what was done. It is confusing with the distance measurements – maybe too much detail?! Please rephrase, put in 1-2 references for the sampling method, and please correct English language, grammar issues (capital letters after comma etc).

Response 1:  Thanks, We put references 21, 22. Cited for the sampling method and rewrote the snail sampling method.

Point 2: Line 131: put in spacing around the bracket “(tissues)”

Response 2: Thanks! We did it.

Point 3: Line 117-120: Ad response 9: There is only a ref 18 in the section on snail sampling. And that ref is put in to support the statement on the infection rates. If it is the same ref, then put it in again when you describe the snail sampling, e.g. in line 120.

Response 3:  This is also the reference citation problem. We changed to the correct ones

Point 4: Line 143: put space in after 3’

Response 4: Thanks, We put space in after 3’

Point 5: Line 159-162: Please rephrase or correct the English language – not clear.

Response 5:  Yes, we modified these sentences here.

Point 6: Line 183: The snail species name has been written earlier so write O. hupensis. Please check throughout main text.

Response 6:  Thanks! We double checked the snail species name and adjusted the same in the whole main text.

Point 7: Table 1: For the Pos. Microscopy results, it shows that there is 1 for Hubei province, but the total shows 2 – how is that possible?? Please correct results – this is very important.

Response 7:  Sorry for that , the Microscopy results should be just one positive case for Hubei Province, so the total should change to 1. And also the positive rate need to change to 0.4%.

Point 8: Line 247 and table 3: It says 28 different laboratories, but there is only 27 mentioned in the table. I guess you removed the last CDC lab from Chongqin Province from the table 2, but please revise text accordingly throughout the manuscript. Is it 27 or 28 labs?

Response 8: Sorry for that. It should be 28 different laboratories except Chongqing, I found out that the Fujian CDC was missing. Now, added it again.

Reviewer 3 Report

The authors had cleared most part of the points. However, some more improvement/clarifications are needed.

Regarding PCR and LAMP: It looks the authors are using the technique described by Kumagai et al. However I could not find the citation in the materials and methods.  

The citations are not correct and need double checking.

The English language should be improved.

What is the meaning to test different stages like mother sporocists or daughter sporocists?

Author Response

Response to Reviewer 3 Comments

Dear Professor,

We appreciate you a lot for your nice comments.

We did a thorough revision of the manuscript, and addressed your comments point by point as followings.

Thanks and regards,

Zhiqiang Qin

Point 1: Regarding PCR and LAMP: It looks the authors are using the technique described by Kumagai et al. However I could not find the citation in the materials and methods.  

Response 1: Yes, you are right, we used the technique described by Kumagai et al. Please see Reference 17,

17.   Kumagai, T.; Furushima-Shimogawara, R.; Ohmae, H.; Wang, T. P.; Lu, S.; Chen, R.; Wen, L.; Ohta, N. Detection of early and single infections of Schistosoma japonicum in the intermediate host snail, Oncomelania hupensis, by PCR and loop-mediated isothermal amplification (LAMP) assay. Am. J. Trop. Med. Hyg. 2010, 83(3), 542-548. doi: 10.4269/ajtmh.2010.10-0016.

Point 2: The citations are not correct and need double checking.

Response 2: We agree with the comments! We double checked whole manuscript and modified the reference citation.

Point 3:The English language should be improved.

Response 3: Thanks! We ask for help for edited by native English Speaking expert before we submitted this time.

Point 4:What is the meaning to test different stages like mother sporocists or daughter sporocists?

Response 4: Good question! Yes, as we know that there are three different satges in the snail mother and daughter sporocist and cercaria.  It is important to know that each of these stages in the snail can be detected.